# Converting copper sulfide to copper with surface sulfur for electrocatalytic alkyne semi-hydrogenation with water

Yongmeng Wu [1,3], Cuibo Liu [1,3], Changhong Wang [1,3], Yifu Yu [1], Yanmei Shi [1] & Bin Zhang [1,2✉]

Electrocatalytic alkyne semi-hydrogenation to alkenes with water as the hydrogen source using a low-cost noble-metal-free catalyst is highly desirable but challenging because of their over-hydrogenation to undesired alkanes. Here, we propose that an ideal catalyst should have the appropriate binding energy with active atomic hydrogen (H*) from water electrolysis and a weaker adsorption with an alkene, thus promoting alkyne semi-hydrogenation and avoiding over-hydrogenation. So, surface sulfur-doped and -adsorbed low-coordinated copper nano-wire sponges are designedly synthesized via in situ electroreduction of copper sulfide and enable electrocatalytic alkyne semi-hydrogenation with over 99% selectivity using water as the hydrogen source, outperforming a copper counterpart without surface sulfur. Sulfur anion-hydrated cation ($S^{2-}$-$K^+$($H_2O$)$_n$) networks between the surface adsorbed $S^{2-}$ and $K^+$ in the KOH electrolyte boost the production of active H* from water electrolysis. And the trace doping of sulfur weakens the alkene adsorption, avoiding over-hydrogenation. Our catalyst also shows wide substrate scopes, up to 99% alkenes selectivity, good reducible groups compatibility, and easily synthesized deuterated alkenes, highlighting the promising potential of this method.

[1] Department of Chemistry, Institute of Molecular Plus, School of Science, Tianjin University, Tianjin, China. [2] Tianjin Key Laboratory of Molecular Optoelectronic Science, Collaborative Innovation Center of Chemical Science and Engineering, Tianjin, China. [3] These authors contributed equally: Yongmeng Wu, Cuibo Liu, Changhong Wang. ✉email: bzhang@tju.edu.cn

Selective hydrogenation of alkynes to alkenes is a fundamental and significant transformation in synthetic chemistry. This transformation is also a classical model reaction to evaluate the performance of a newly developed catalyst or hydrogenation method[1–8]. Although significant progress has been achieved, current reports are still mainly relying on the expensive noble-metal catalysts (e.g., Pt, Pd, Ru, or their related alloys)[9–13] or complicated metal complexes[14–17] with gaseous hydrogen ($H_2$) or expensive and/or toxic organic hydrogen sources, causing severe concerns on the cost, safety, and sustainability. To solve these problems, an electrochemical strategy has been recently developed by our group for selective semi-hydrogenation of alkynes over a Pd-P cathode by using $H_2O$ as the hydrogen source[18]. However, good alkenes selectivity required the accurate control of the applied potential and reaction time. So, time-prolonged or potential-sensitive over-hydrogenation of alkynes to alkanes is still a big obstacle for the practical synthesis of high-purity alkenes[19,20]. At present, a catalyst that can intrinsically govern the semi-hydrogenation of alkynes to alkenes is still lacking. Thus, exploring a non-noble-metal-nanostructured electrocatalyst for highly selective alkyne semi-hydrogenation in a wide range of potential with time-independent selectivity is urgently demanded.

Electrochemistry has provided an efficient and sustainable platform for synthesizing value-added fine chemicals[21–26]. Cu-based catalysts are widely used in many electrochemical reactions due to their abundance, robustness, and superior catalytic activity[27–34]. As the hydrogen adsorption free energy ($\Delta G_{H*}$) is relatively large[27], Cu materials can inhibit the competing hydrogen evolution reaction (HER), and thus serve as the ideal candidates for electroreduction reactions (e.g., carbon dioxide reduction reaction ($CO_2$ RR)[28–31], halide deuteration[33], and nitrate RR[34]). Doping and surface modification are advantageous to high carbon products of $CO_2$ reduction by controlling the adsorption of intermediates[29,30]. In fact, the electrode materials often experience structural reconstruction, including the changes in morphology, composition, and crystal face under the electrochemical conditions. For instance, copper oxide-derived nanostructures exhibited an improved performance for $CO_2$ RR for rich grain boundaries, exposed high-index facets, and low-coordinated Cu[34]. Metal chalcogenides and MoNi alloy reconstructed to produce surface-adsorbed chalcogenates[35] and $Mo_2O_7{}^{2-}$ ions[36], both of which were proven to accelerate water electrolysis. In addition, surface $S^{\delta-}$ adsorbed inert Au or In would promoted water electrolysis to generate the active atomic hydrogen (H*)[37,38]. Furthermore, doping S into Pd could weaken the adsorption of alkenes to inhibit its over-hydrogenation in the $H_2$-participated thermal catalysis[39] and the semi-hydrogenation of alkynes to alkenes has also been achieved by homogeneous and heterogeneous Cu catalysis under high pressure of $H_2$ at elevated temperature[40–42]. In these regards, we speculated that sulfur-modified Cu nanostructures, especially synthesized via in situ electroreduction of self-supported precursors, might be highly efficient for electrocatalytic semi-hydrogenation of alkynes. However, in situ electrochemically formed Cu nanostructures with surface sulfur have not been explored for selective semi-hydrogenation of alkynes using $H_2O$ as the hydrogen source.

Here, surface sulfur-doped and -adsorbed Cu nanowire sponges (Cu-S NSs) were designedly synthesized by in situ electroreduction of CuS nanowire arrays/Cu foam (CuS NAs). Cu-S NSs were highly active for electrochemical transfer semi-hydrogenation of alkynes to alkenes with up to 99% selectivity, broad substrate scope, and good functional group compatibility by using $H_2O$ as the hydrogen source (Fig. 1). Electrochemical in situ X-ray diffraction (XRD), Raman spectroscopy, and X-ray absorption spectroscopy (XAS) demonstrated the conversion process from CuS NAs to Cu-S NSs. Both experimental and computational results revealed the crucial roles of surface-adsorbed and lattice-doped sulfur in Cu-S NSs for selective alkyne semi-hydrogenation. This sulfur-promoted method was also suitable for the hydrodehalogenation of aryl iodide.

## Results

**Materials synthesis and characterizations.** The self-supported CuS NAs were synthesized by a facile sulfuration of $Cu(OH)_2$ nanowire precursors (Fig. 2a and Supplementary Figs. 1a, b). The scanning electron microscopy (SEM) and transmission electron microscopy (TEM) images (Fig. 2d and Supplementary Figs. 2a–c) showed that CuS nanowires assembled by small nanoparticles were directly grown on a copper foam (CF). The lattice fringes with an interplanar distance of 0.305 nm in the high-resolution TEM (HRTEM) image (Supplementary Fig. 2d) were attributed to the (102) planes of CuS[42]. The energy-dispersive X-ray mapping image (Supplementary Fig. 2e) implied that Cu and sulfur were uniformly distributed throughout the entire nanostructure. All the diffraction peaks in the XRD pattern (Fig. 2c) could be indexed to the hexagonal structure of CuS (JCPDS No. 06-0464). The prominent peaks at 932.8 eV (Cu $2p_{3/2}$) and 952.8 eV (Cu $2p_{1/2}$) in the X-ray photoelectron spectra belonged to $Cu^{2+}$, whereas the characteristic signals at 163.5 eV (S $2p_{3/2}$) and 162.3 eV (S $2p_{1/2}$) were assigned to $S^{2-}$ (Supplementary Figs. 2e and 3)[43]. These results suggested the successful synthesis of CuS NAs.

The as-prepared CuS NAs were then transformed into Cu-S NSs via an electrochemical reduction in 1.0 M KOH with a scan rate of 5 mV s$^{-1}$ (Fig. 2a). The linear sweep voltammetry (LSV) curves showed three typical reduction peaks at −0.8, −1.05, and −1.3 V vs. Hg/HgO corresponding to the reduction of $CuS/Cu_8S_5$, $Cu_8S_5/Cu_2S$, and $Cu_2S/Cu$, respectively. These reduction peaks first appeared, gradually faded, and finally vanished at the fourth LSV scan, indicating the rapidly complete reduction of CuS precursors (Fig. 2b). The details will be discussed later. This result is consistent with the Pourbaix diagram of Cu-S system (Supplementary Fig. 4), in which CuS is not thermodynamically stable in aqueous solutions at pH 14, and can become metallic Cu at potentials more negative than −0.8 V vs. standard hydrogen electrode (−1.3 V vs. Hg/HgO)[44,45]. The electroreduction-induced sulfur stripping could decrease the size of the nanowires and cause the transformation of NAs to NSs (Fig. 2d and Supplementary Figs. 5a–c). The XRD pattern showed the disappearance of CuS and the presence of $Cu_2O$ after the electroreduction (Fig. 2c), which might be owing to the inevitable oxidation of the sample in air. The clear fringes with a lattice spacing of 0.24 nm in the HRTEM image also confirmed the $Cu_2O$ phase (Supplementary Fig. 5b). The energy-dispersive X-ray spectroscopy (EDS) data suggested the sulfur content of the Cu-S NSs was about 5.5%, implying the severe leaching of sulfur during the reduction process and the existence of trace sulfur in the in situ-formed Cu-S NSs (Supplementary Fig. 6). X-ray photoelectron spectroscopy (XPS) was also used to investigate the valence state of Cu-S NSs. After the electroreduction conversion, the atomic percentage of sulfur decreased from 42% to 4.6%, in accordance with the EDS results. The peaks at 163 eV (S $2p_{3/2}$) and 162 eV (S $2p_{1/2}$) were assigned to $S^{2-}$. Further, a new peak at 168 eV was derived from the oxidation of surface sulfur species in air[44]. In Cu spectra, the peaks at 932.8 and 953.8 eV were assigned to the $Cu^+$ $2p_{3/2}$ and $Cu^+$ $2p_{1/2}$ of $Cu_2O$, whereas the small peaks at 954.1 and 934.6 eV with the satellite peak at 942 eV correspond to the $Cu^{2+}$ of CuO because of its oxidation in air[46]. These results indicated that CuS NAs underwent morphological and structural evolution during the electrochemical reduction process.

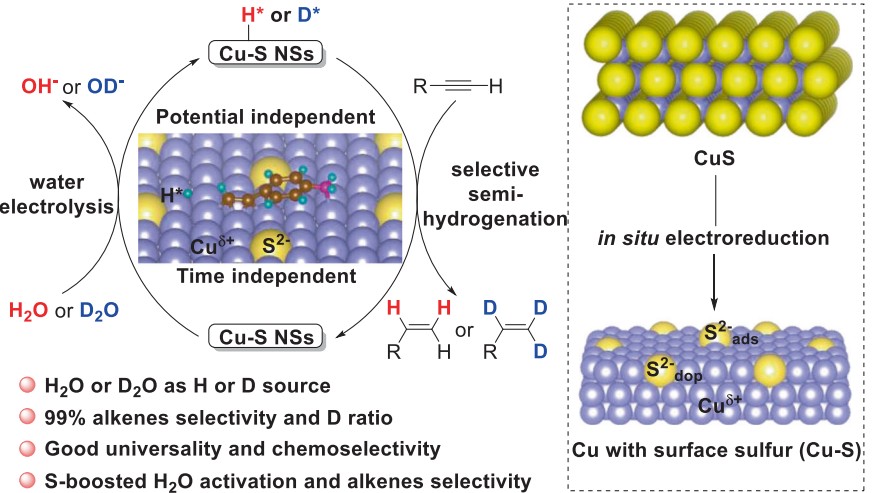

**Fig. 1 A schematic illustration of the semi-hydrogenation strategy.** Electrocatalytic transfer semi-hydrogenation of alkynes with $H_2O$ to corresponding alkenes over in situ-formed Cu with surface sulfur doping and adsorption (Cu-S).

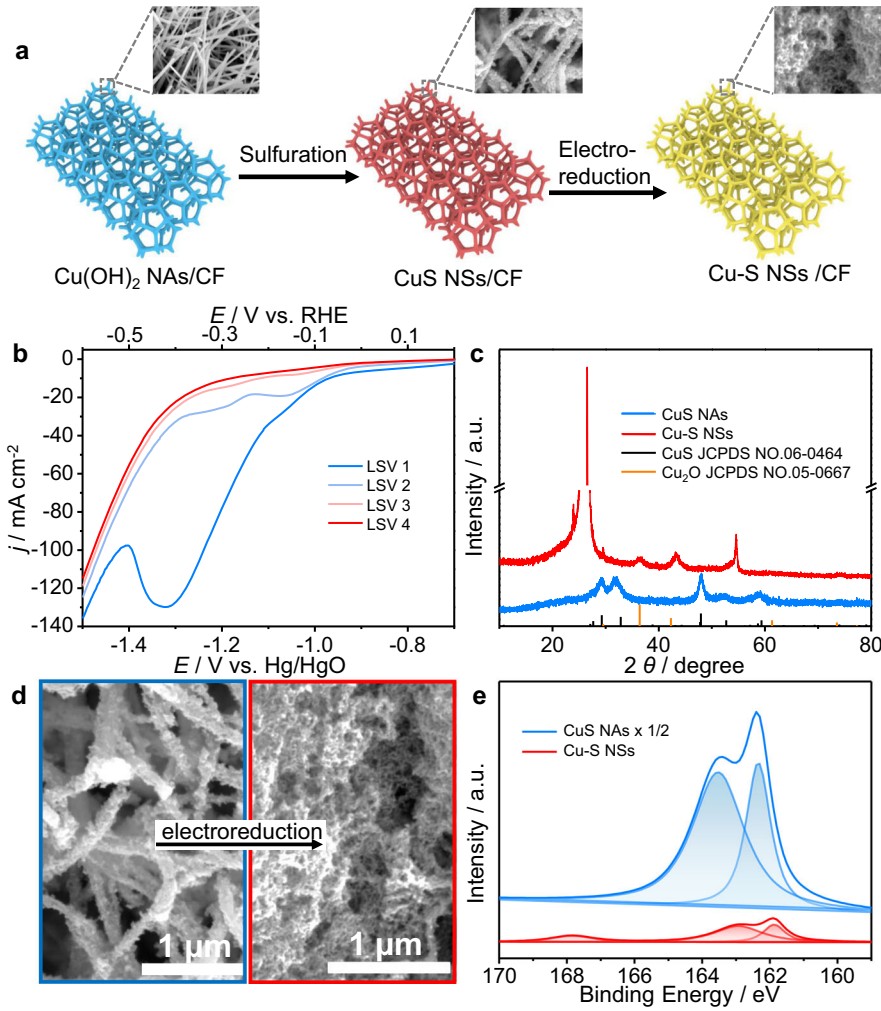

**Fig. 2 Converting CuS NAs to Cu-S NSs under cathodic potential. a** Scheme illustrating the in situ electroreductive conversion of CuS NAs precursors to Cu-S NSs. **b** LSV curves in the electroreduction process of CuS NAs. **c** XRD patterns, **d** SEM images, and **e** S 2$p$ spectra of CuS NAs and in situ-formed Cu-S NSs.

X-ray spectroscopy (EDS) data suggested the sulfur content of the Cu-S NSs was about 5.5%, implying the severe leaching of sulfur during the reduction process and the existence of trace sulfur in the in situ-formed Cu-S NSs (Supplementary Fig. 6). XPS was also used to investigate the valence state of Cu-S NSs. After the electroreduction conversion, the atomic percentage of sulfur decreased from 42% to 4.6%, in accordance with the EDS results. The peaks at 163 eV (S $2p_{3/2}$) and 162 eV (S $2p_{1/2}$) were assigned to $S^{2-}$. Then, a new peak at 168 eV was derived from the oxidation of surface sulfur species in air[44]. In Cu spectra, the peaks at 932.8 and 953.8 eV were assigned to the $Cu^+$ $2p_{3/2}$ and $Cu^+$ $2p_{1/2}$ of $Cu_2O$, whereas the small peaks at 954.1 and 934.6 eV with the satellite peak at 942 eV correspond to the $Cu^{2+}$ of CuO because of its oxidation in air[46]. These results indicated that CuS NAs underwent morphological and structural evolution during the electrochemical reduction process.

Furthermore, a series of in situ characterizations (the following characterizations were performed in situ unless otherwise stated) were conducted to study the structural evolution of CuS under the electrochemical condition. First, potential-dependent XRD was used to monitor the phase transformation of the sample (Fig. 3a). To exclude the substrate interference, CuS powders were first stripped off from Cu foam substrate and loaded on carbon fiber paper (CP) for XRD tests. The peak at 54.5° was one of the characteristic peaks of CP (JCPDS No. 26–1076). The XRD pattern of the initial sample showed two peaks at 48.1° and 59.2°, corresponding to the (110) and (116) crystal face of the covellite CuS phase. When the applied potential was decreased, the above two peaks disappeared and a new peak at 47.5° probably belonged to $Cu_8S_5$ (113) raised at −0.85 V vs. Hg/HgO. At −1.0 V vs. Hg/HgO, three characteristic peaks of the hexagonal $Cu_2S$ phase at 46.0°, 48.3°, and 53.9° became prominent at the expense of the $Cu_8S_5$. At −1.2 V vs. Hg/HgO, two characteristic peaks at 43.3° and 50.6° of cubic Cu appeared, corresponding to the (111) and (200) crystal facets. With the potential decreasing, the two new peaks became stronger and the $Cu_2S$ peaks vanished ultimately.

Thus, these XRD analyses suggested the reduced conversion process of CuS → $Cu_8S_5$ → $Cu_2S$ → Cu. No peaks from CuO or $Cu_2O$ were observed in these XRD patterns, indicating the detection of $Cu_2O$ and CuO in the ex situ XRD and XPS was due to the oxidation of metal Cu through exposure to air. As a result, metal Cu with trace surface sulfur was the final stable phase for the in situ electroreduction of CuS NAs, although its easy oxidation in air.

The conversion process from CuS NAs to Cu-S NSs was also confirmed by electrochemical Raman spectroscopy. Figure 3b depicted that the S–S bond vibration of CuS at 472 cm$^{-1}$ decreased gradually with decreasing the applied potential[43] and completely disappeared at −1.3 V vs. Hg/HgO, reflecting that the detected surface sulfur in EDS and XPS was isolated in Cu-S NSs. The isolated surface sulfur indicated the S-doping and -adsorption on the surface of in situ-formed Cu. Meanwhile, the background fluorescent intensity of Cu ascended with the decrease of the applied potential, further accounting for converting CuS to Cu. Thus, the Raman data confirmed the complete conversion of CuS to Cu and the surface S-doping and adsorption on in situ-formed Cu.

To understand the real electronic configuration and local coordination environment of in situ-formed Cu-S NSs, electrochemical XAS was performed. The Cu $K$-edge extended X-ray absorption near-edge structure spectra (XANES) and the corresponding derivative curves of Cu-S NSs at −1.3 V vs. Hg/HgO both exhibited similar features to Cu foil (Fig. 3c and Supplementary Fig. 7), consistent with the XRD results in Fig. 3a[47]. However, the absorption edge position of Cu-S NSs was located between those of $Cu_2S$ and Cu, suggesting Cu-S was more oxidized than Cu foil (inset in Fig. 3c). This might be ascribed to the existence of low-coordinated unsaturated Cu, which was caused by the stripping of sulfur in electrochemical reduction. The local coordination environment was further analyzed by the Fourier-transformed extended X-ray absorption fine structure curves (Fig. 3d). Under the applied potential of −1.3 V vs.

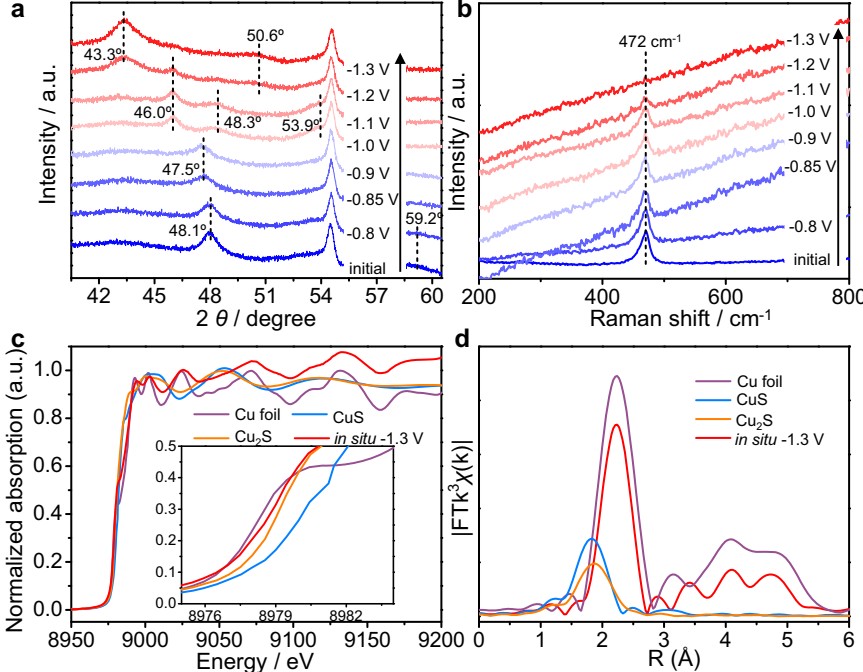

**Fig. 3 Electrochemical in situ characterization of converting CuS NAs to Cu-S NSs under cathodic potential. a** In situ potential-dependent XRD pattern and **b** in situ potential-dependent Raman spectra of CuS NAs. **c** In situ XANES and **d** EXAFS spectra of CuS NAs at −1.3 V vs. Hg/HgO. The spectra of Cu foil, CuS, and $Cu_2S$ were also shown for comparison.

Hg/HgO, the sample mainly consisted of Cu-Cu bonds with a bond length of 2.3 Å and a small amount of Cu-S bond with a bond length of 1.8 Å. The latter further confirmed the presence of surface sulfur in in situ-formed samples. The simulated coordination number of as-formed Cu-S was lower than that of Cu foil. Thus, the in situ-formed Cu-S NSs owned more low-coordinated Cu atoms, as observed in oxide-derived Cu[34]. In addition, the absence of the Cu-O bond in XAS spectra confirmed that the $Cu_2O$ phase of the XRD pattern (Fig. 2c) originated from the oxidation of activated Cu-S NSs in air. This hypothesis was further proved by the quickly complete conversion from Cu-S into $Cu_2O$ after exposing the fresh Cu-S NSs to air without bias, as shown in the XAS results (Supplementary Fig. 8). Thus, in situ-formed Cu-S NSs were very unstable in air and should be directly used as the electrocatalyst without air exposure.

The above ex situ and in situ results and discussions demonstrated that the rationally designed in situ electroduction of CuS NAs could successfully synthesize low-coordinated Cu with surface sulfur doping and adsorption as our target materials. This in situ electroreduction strategy of CuS precursors might be an important addition to the toolbox of chemical conversion synthesis[48].

**Electrocatalytic semi-hydrogenation of alkynes with $H_2O$ over Cu-S NSs.** After Cu-S NSs were in situ synthesized in a divided three-electrode system (Supplementary Fig. 9), 4-ethynylaniline (**1a**), the model substrate, was rapidly added to the 1.0 M KOH electrolyte containing 1, 4-dioxane (Diox, $V_{Diox}:V_{Water} = 2:5$) as the co-solvents at a constant potential for evaluating the electrocatalytic semi-hydrogenation of alkynes. The LSV curves displayed that the current density increased slightly between −1.15 and −1.3 V vs. Hg/HgO after adding **1a**, implying the easier reduction of **1a** than HER in this potential range. At below −1.3 V vs. Hg/HgO, the two LSV curves almost overlapped, and the increase in current density might mainly be ascribed to the HER (Supplementary Fig. 10). Impressively, over 99% selectivity

of product alkene **2a** was observed from −1.25 to −1.5 V vs. Hg/HgO (Fig. 4a), suggesting this potential-independent selectivity over Cu-S NSs. From −1.3 to −1.4 V vs. Hg/HgO, 99% conversion yields were achieved with the complete consumption of **1a** within 4.0 h. Time-dependent curves showed that **2a** selectivity did not vary with reaction time, even after the complete disappearance of **1a** (Fig. 4b).

These satisfactory results demonstrated the highly selective semi-hydrogenation of alkynes to alkenes was governed by the Cu-S NSs cathode itself, which was hardly impossible to the current reports. As expected, the inherent selectivity of the catalyst enabled the reaction to maintain excellent alkenes selectivity (>98%) even under constant current conditions (Supplementary Fig. 11), further demonstrating the excellent alkene selectivity was intrinsically controlled by the Cu-S NSs rather than the input potential/current or the reaction time. In addition, the conversion yield and selectivity of **2a** were much worse when employing in situ-formed Cu by the electroreduction of $Cu(OH)_2$ NAs as the cathode (Fig. 4c). In particular, no **2a** was detected from −1.3 to −1.4 V vs. Hg/HgO as we expected, further demonstrating the unique advantage of our Cu-S NSs in controlling the selectivity of alkenes. Furthermore, in situ-formed Cu-S NSs cathode was highly active and durable for this selective hydrogenation (Fig. 4d), implying good durability. Deuterium-labeling experiments verified $H_2O$ was the sole hydrogen source for this electrochemical semi-hydrogenation of alkynes to alkenes (Supplementary Fig. 21). Thus, the excellent selectivity and activity for the semi-hydrogenation of alkynes with water as the hydrogen source were potential-independent and time-independent over the in situ-formed Cu-S NSs cathode, holding promising potential for practical and long-term alkene synthesis.

**Mechanistic insight into surface sulfur promotion.** The high activity and selectivity origin of this Cu-S NS-mediated electrocatalytic semi-hydrogenation of alkynes to alkenes was further

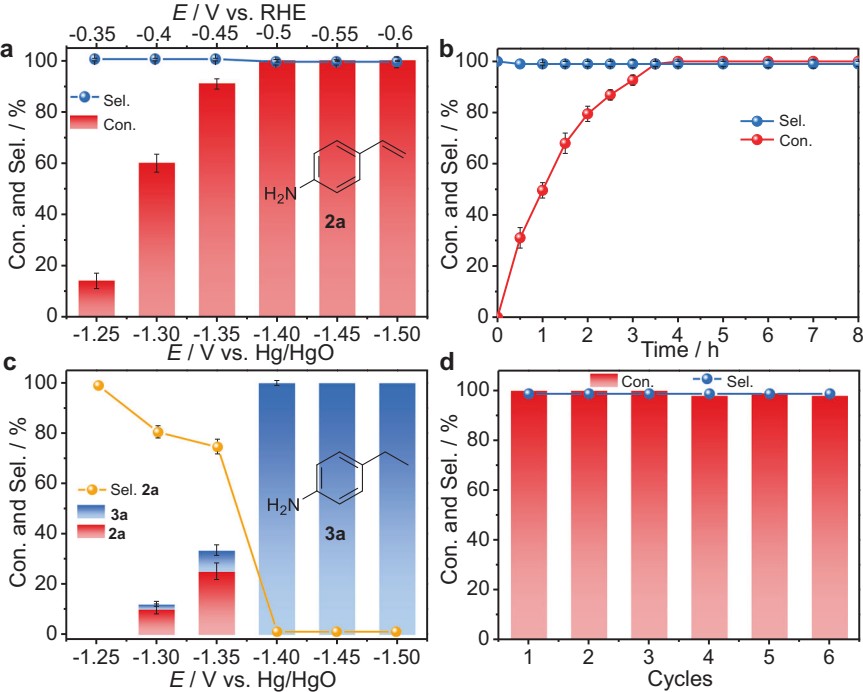

**Fig. 4 Performances of in situ-formed Cu-S NSs for electrocatalytic semi-hydrogenation of an alkyne. a** Potential-dependent and **b** time-dependent **1a** conversion yield (Con.) and **2a** selectivity (Sel.) over Cu-S NSs. **c** Potential-dependent **1a** Con. and **2a** Sel. over Cu NAs. **d** Cycle-dependent **1a** Con. and **2a** Sel. over Cu-S NSs. Error bars correspond to the SD of three independent measurements.

explored. The introduction of $S^{\delta-}$ to inert metal (e.g., Au, In) was reported to promote the activation of water and finally boost the activity and efficiency of $CO_2$ RR[37,38]. Then, the role of surface sulfur in our reactions was investigated. First, Cu nanowire arrays (donate as Cu NAs) were synthesized by in situ electroreduction of $Cu(OH)_2$ NAs. After in situ forming Cu NAs, we added $Na_2S$ (0.05 mmol) to the solution and obtained Cu-$S_{add}$ NAs. A remarkable advance of the onset potential from the LSV curves was observed after introducing $Na_2S$, indicating sulfur-promoted HER (Fig. 5a). When we added a small amount of $Zn^{2+}$ to block the surface $S^{2-}$ sites through the strong interaction between $Zn^{2+}$ and $S^{2-}$, the HER activity of Cu-$S_{add}$ NAs was obviously reduced, confirming the enhancement of surface $S^{2-}$ for HER (Fig. 5a). However, the HER performance for Cu-$S_{add}$ after surface $S^{2-}$ blocked was still better than Cu NAs. We speculated the surface $S^{2-}$ might diffuse and dope into the lattice, as indicated in the isolated Cu-S bond in the Raman spectra of Cu-S NSs (Fig. 3b). Furthermore, the alkyne **1a** conversion and alkene **2a** selectivity over Cu-$S_{add}$ NAs were 99% and 99%, obviously higher than 21% and 81% over corresponding $Cu(OH)_2$-derived Cu without surface sulfur, respectively. Impressively, Cu-$S_{add}$ NAs demonstrated comparable performance with Cu-S NSs, suggesting the addition of $Na_2S$ could effectively modulate the activity and selectivity of the alkynes semi-hydrogenation. However, after removing the surface-adsorbed $S^{2-}$ by $Zn^{2+}$ (denoted as Cu-$S_{add}$-$Zn^{2+}$ NAs), the conversion of **2a** decreased clearly, but the **2a** selectivity was almost unchanged. These observations illustrated that the surface-adsorbed sulfur promoted the generation of active H* from water electrolysis and the $S^{2-}$ doping into the Cu lattice greatly enhanced the alkenes selectivity, jointly determining the highly efficient semi-hydrogenation of alkynes to alkenes over Cu-S NSs. Indeed, the kinetic isotope effect with the value of 3.2 ($k_H/k_D$) at $-1.3$ V vs. Hg/HgO in the Cu-S NSs system was indicative of the dissociation of $H_2O$ as the rate-determining step for the semi-hydrogenation of alkynes to alkenes (Supplementary

Fig. 12). Thus, it was possibly deduced that surface-adsorbed sulfur could accelerate the production of H* via water electrolysis and subsequently promote the hydrogenation of alkynes to alkenes.

Recent studies have demonstrated that the surface-adsorbed anions (e.g., $S^{\delta-}$, $Cl^-$, $F^-$) exerted a vital role in accelerating the activation of $H_2O$ in alkaline media[37,38]. The anion-hydrated cation networks ($S^{\delta-}$-$K^+(H_2O)_n$) could be formed in the double layer structure through the non-covalent Coulomb interaction between the surface anion and the hydrated cation. The proposed cation/anion pair complex facilitated the dissociation of $H_2O$ to form active H* species, which was usually a slow step and restrained the HER via the Volmer step ($2H_2O + M + 2e^- \rightarrow 2M$-H* $+ 2OH^-$). Therefore, control experiments were done to verify whether a similar promoting effect was involved in our hydrogenated reaction. When 1.0 M tetramethylammonium hydroxide solution was used instead of KOH electrolyte, Cu-S NSs exhibited much poor HER activity and sluggish alkynes hydrogenation rate at $-1.3$ V vs. Hg/HgO under identified conditions (Figs. 5c, d), which might be due to the weaker interaction between $S^{2-}$ and $TMA^+$ than hydrated $K^+$, and hence more inferior capability to promote $H_2O$ dissociation. By changing the cation from $K^+$ to $Na^+$, slight decreases for both HER and **1a** transformation were observed (Figs. 5c, d). We rationalized the better performance for $K^+$ might be due to the smaller ionic hydration number and radius of the hydrated cation of $K^+(H_2O)_n$ ($n = 7$ for $K^+$ vs. 13 for $Na^+$)[49] that resulted in stronger interactions with $S^{2-}$ on Cu surface, thus promoting the water activation. Furthermore, for $Cu(OH)_2$-derived Cu without surface sulfur, we could not observe noticeable differences in the HER activity after adding different cations (Supplementary Fig. 13). These results revealed that the surface-adsorbed $S^{2-}$ on Cu enhanced the alkyne semi-hydrogenation activity by promoting $H_2O$ activation via the interaction of $S^{2-}$ with hydrated cations.

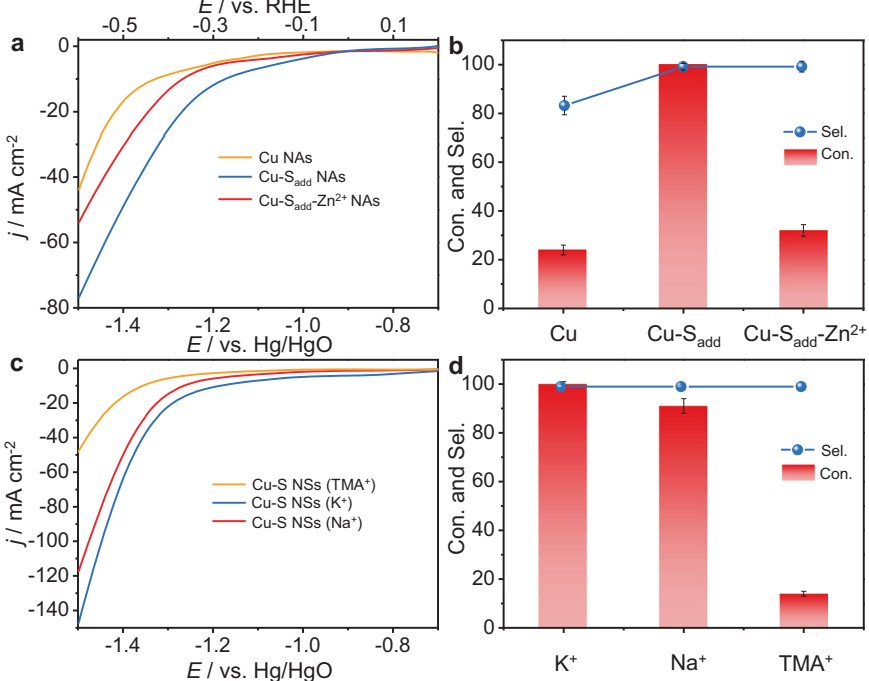

**Fig. 5 The effect of sulfur ions and cations in the electrolyte. a** LSV curves recorded in 1.0 M KOH without **1a**, and **b** **1a** Con. and **2a** Sel. obtained with 0.1 mmol of **1a** over different cathodes. **c** LSV curves recorded in 1.0 M MOH (M = $Na^+$, $K^+$, and $TMA^+$) electrolyte without **1a** over Cu-S NSs cathode. **d** **1a** Con. and **2a** Sel. obtained in 1.0 M MOH (M = $Na^+$, $K^+$, and $TMA^+$) electrolyte with **1a** over Cu-S NSs cathode. Error bars correspond to the SD of three independent measurements.

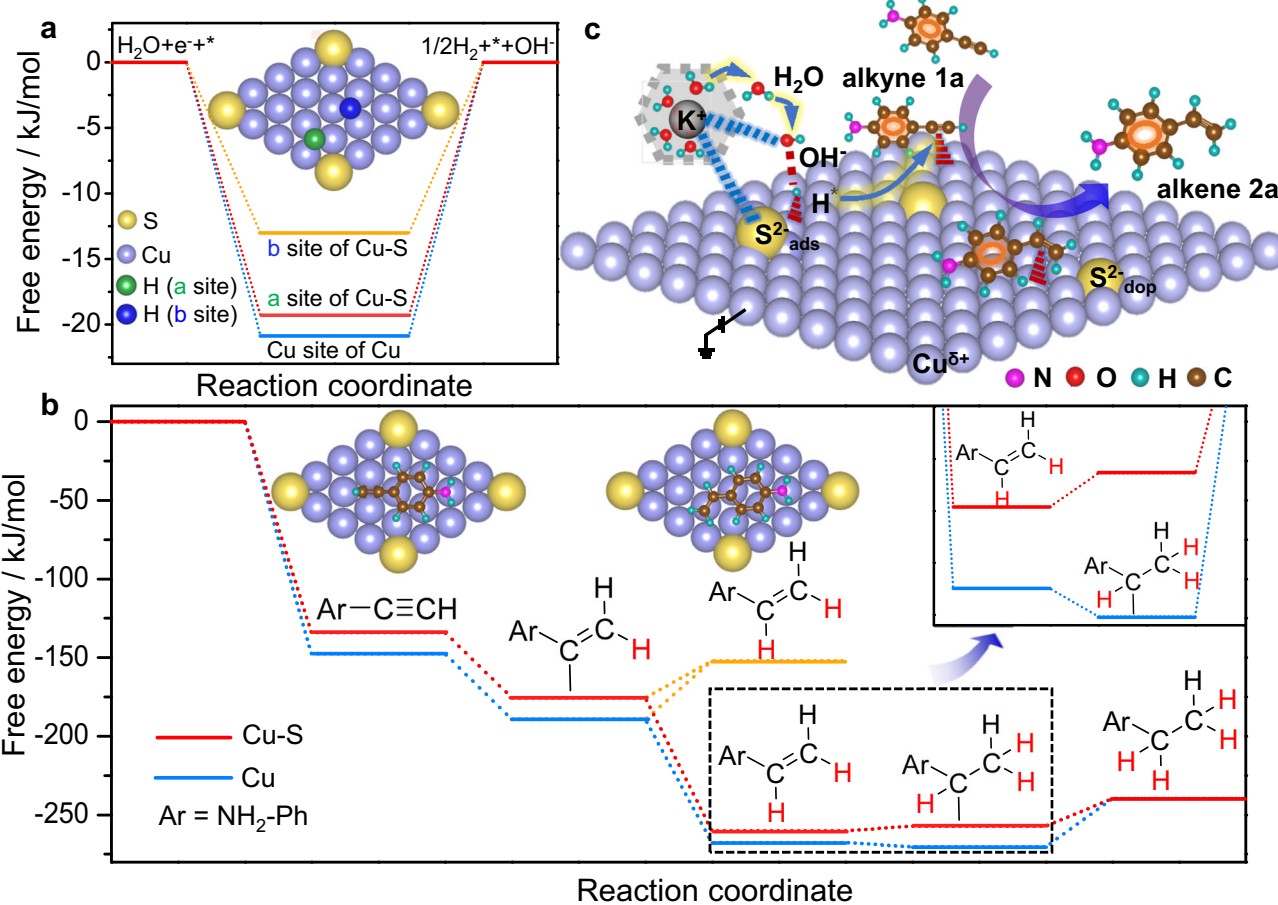

**Fig. 6 Theoretically understanding the activity and selectivity origin of Cu-S toward electrocatalytic semi-hydrogenation of an alkyne. a** The calculated free energy diagrams for hydrogen adsorption and **b** free-energy diagram for alkyne semi-hydrogenation reaction over Cu and Cu-S. **c** The proposed reaction mechanism for alkyne semi-hydrogenation reaction over Cu-S cathode.

Density functional theory (DFT) calculations were performed to fundamentally understand the promoting origin of surface sulfur in this electrochemical semi-hydrogenation of alkynes to alkenes. We first calculated the reaction Gibbs free energies ($\Delta G$) for the HER on pure Cu and sulfur-doped and -adsorbed Cu-S NSs. The $\Delta G$ values of H* formation on Cu atoms near and far from S atoms in Cu-S were −19.3 and −12.54 kJ/mol, respectively, which were more positive than the value of −21.23 kJ/mol for pure Cu (Fig. 6a). The relatively weak H* affinity of Cu-S was favorable for the subsequent semi-hydrogenation of the alkynes, accounting for the experiment results. In addition, the free energy of alkene **2a** on Cu-S was positive than that on pure Cu, indicating that **2a** was readily desorbed from Cu-S (Fig. 6b). We have also compared the $\Delta G$ for further hydrogenation of **2a** by adding another H over Cu-S and pure Cu, respectively. The $\Delta G$ of +3.13 kJ/mol suggested that over-hydrogenation of alkenes to alkanes was thermodynamically unfavorable for Cu-S NSs, implying the intrinsically high selectivity of alkenes over Cu-S NSs (Fig. 6b). In fact, **2a** could be completely hydrogenated to the corresponding alkanes over Cu(OH)₂-derived Cu, but **2a** could not be further hydrogenated to alkane over Cu-S NSs, supporting the theoretical discussion (Supplementary Fig. 14). Therefore, a possible reaction mechanism was proposed in Fig. 6c (**1a** as the example). A $S^{\delta-}$-K⁺(H₂O)$_n$ network was formed via the interaction between the surface $S^{\delta-}$ and hydrated K⁺. The near-surface water molecules were facilely activated to produce the adsorbed H* species on the Cu atom close to S. In our alkaline system (pH = 13.6), the proton-coupled

electron transfer process might be not involved for the semi-hydrogenation of alkynes because of the extremely low concentration of proton[50]. We thus reasoned an addition pathway of H* to alkynes accounting for the formation of alkene products. Subsequently, H* was transferred to **1a**, which was activated and adsorbed on a nearby Cu atom, and then coupled with another H* to generate the alkene product **2a**. Because of the weak adsorption of **2a** on Cu-S and the thermodynamic constraints for further hydrogenation, **2a** desorbed with the regeneration of the Cu-S surface for the next cycle.

**The universality of the electrocatalytic semi-hydrogenation of alkynes with H₂O over Cu-S NSs.** The general applicability of this electrochemical semi-hydrogenation over Cu-S NSs was tested (Table 1). A series of terminal alkynes with both electron-withdrawing and -donating substituents on the aryl ring reacted well to deliver the corresponding alkenes with excellent selectivity (99%) and good to high conversion yields (**2a** to **2i**). In particular, the easily reducible C-Cl, C-Br, and C=O bonds, which were usually challenging by traditional methods, retained well during the electrochemical process, providing good chances for assembling more complex molecules (**2d**–**f**). The -NH₂, -COOH, pyridyl, and thienyl groups, which usually posed deactivation to the metal catalysts, had nearly no influence on the activity and product selectivity, demonstrating the robust property of our Cu-S NSs cathode (**2a**, **2g**, **2i**–**j**). Interestingly, by replacing H₂O with

**Table 1 Substrate scope for electrocatalytic transfer semi-hydrogenation of alkynes with $H_2O$ over Cu-S NSs cathode[a].**

$^a$Reaction conditions: alkyne substrates (0.1 mmol), Cu-S NSs (working area: 1.0 cm$^2$), 1.0 M KOH (Diox/H$_2$O, 2:5 v/v, 7 mL), room temperature, −1.3 V vs. Hg/HgO. Conversion yields were reported, and the data in parentheses were the alkenes selectivity.
$^b$0.5 M K$_2$CO$_3$ was used as the electrolyte using Diox/D$_2$O (2:5 v/v, 7 mL) as co-solvents.

$D_2O$, our method was readily applied to the selective synthesis of deuterated alkenes with high yields, selectivity, and deuterated efficiency (99%), which could serve as important building blocks for synthesizing deuterated drug molecules, advanced functional materials, or probe molecules for studying reaction mechanism[51,52].

As discussed in the mechanism section above, the introduction of Na$_2$S to in situ-formed Cu could boost the alkyne semi-hydrogenation. We thus check the universality of the sulfur-modified strategy in other hydrogenation reactions. The hydro-dehalogenation was selected as the model reaction due to its essential role in organic synthesis[32]. As expected, an obvious improvement of the electrochemical conversion of 2-iodobenzyl alcohol to benzyl alcohol was observed after adding 0.05 mmol of S$^{2−}$ into the system by employing in situ-formed Cu NAs as the cathode (Fig. 7), further demonstrating the general applicability of this sulfur-treating method.

## Discussion

In summary, the boosted active hydrogen species (H*) and the weak alkenes adsorption were proposed to achieve electrocatalytic alkyne semi-hydrogenation with high activity and selectivity. These considerations made sulfur-doped and -adsorbed low-coordinated Cu an ideal cathode for realizing potential- and time-independent selective alkyne semi-hydrogenation using water as the hydrogen source. Then, the in situ electroreduction of CuS NA precursors were developed to prepare Cu-S NSs. A combined ex situ and in situ result demonstrated the conversion process of CuS → Cu$_8$S$_5$ → Cu$_2$S → low-coordinated Cu with surface-doped and -adsorbed sulfur (Cu-S NSs). Experimental and theoretical results suggested the surface-adsorbed sulfur on Cu-S NSs could interact with the hydrated K$^+$ to form the S$^{δ−}$-K$^+$(H$_2$O)$_n$ network, contributing to the dissociation of H$_2$O to adsorbed active H* species. The lattice-doped sulfur played a decisive role in controlling the high selectivity of alkenes by weakening alkene

adsorption. Furthermore, surface sulfur also made the further hydrogenation of alkene thermodynamically unfavorable, which was another key factor to determine the intrinsically high selectivity of alkenes over Cu-S NSs. This method showed good universality to various substrates and the easily fragile functional groups (e.g., C-Br, C=O) could retain well. In addition, the deuterated alkenes with high yield, selectivity, and deuterated ration rate were easily synthesized. Furthermore, the surface sulfur modification method could also promote electrochemical hydrodehalogenation, illustrating the methodology generality. Our work not only provides an understanding of the activity and selectivity origin of sulfur-modified transition metal toward highly selective semi-hydrogenation of alkynes but also offers a paradigm for designing and synthesizing low-cost, nanostructured electrode via an in situ electrochemical treatment for other electrocatalytic hydrogenation reactions.

## Methods

**Synthesis of self-supported Cu(OH)$_2$ NAs.** Cu(OH)$_2$ NAs were synthesized with slight modification according to the previous report[53]. The commercial CF was cut into a rectangular shape with the size of $1.0 \times 3.0$ cm$^2$. Then the small pieces of CF were carefully washed with 3.0 M acid, acetone, and deionized (DI) water, respectively. Cu(OH)$_2$ NAs supported on CF were self-grown by simple oxidization of the Cu substrate in an alkaline environment. NaOH (3.0 g) and 0.68 g (NH$_4$)$_2$S$_2$O$_8$ were dissolved in 30 mL DI water under vigorous stirring and the precleaned Cu substrate was immersed in it. After 20 min, a blue hydroxide layer was observed on the surface of Cu. The Cu substrate covered with nanowires was taken out from the solution and repeatedly rinsed with DI water, and then dried at room temperature.

**Synthesis of self-supported CuS NAs.** CuS NAs were prepared by the reported hydrothermal sulfidation method[53]. Thiourea (0.22 g) was dissolved into 30 mL ethylene glycol under magnetic stirring at room temperature. Then, the solution was loaded into a 50 ml Teflon-lined autoclave and a piece of freshly treated CF was also added into the autoclave, sealed, heated to 80 °C, and kept at this temperature for 60 min. After the reaction cooled down naturally, the CF with products on it was taken out, washed with water, and dried naturally.

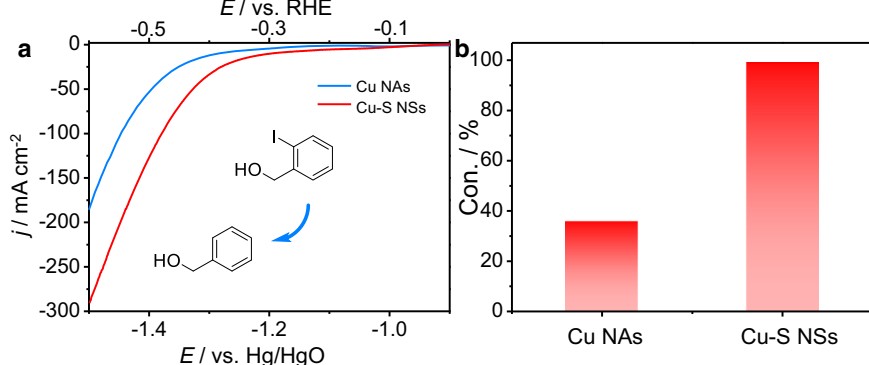

**Fig. 7 Performances of Cu NAs with and without adding Na$_2$S for the hydrodehalogenation of 2-iodobenzyl alcohol. a** LSV curves of Cu NAs with and without adding Na$_2$S for the hydrodehalogenation of 2-iodobenzyl alcohol in 1.0 M KOH. **b** Con. of 2-iodobenzyl alcohol over Cu NAs cathode with and without S$^{2-}$, respectively.

**Synthesis of Cu-S NSs via in situ electroreduction of CuS NAs**. The electro-reduction of Cu-S NNs was conducted on an Ivium-n-Stat electrochemical workstation (Ivium Technologies B.V.) in a typical three-electrode system in 1.0 M KOH. A Hg/HgO with 1.0 M KOH as the inner reference electrolyte was used as the reference electrode. A carbon rod was used as the counter-electrode. CuS NAs/CF was in advance sealed with epoxy, to ensure the exposed area of 1.0 cm$^2$, and then used as the working electrode. The LSV was recorded in the voltage range $-0.7 \sim -1.5$ V vs. Hg/HgO at the scan rate of 5 mV s$^{-1}$ until the reductive peaks disappeared.

**Synthesis of Cu NAs via in situ electroreduction of Cu(OH)$_2$ NAs**. Cu(OH)$_2$-derived Cu NAs were synthesized by a similar in situ electroreduction strategy to those of Cu-S NSs, where CuS NAs were replaced by Cu(OH)$_2$ NAs.

**Characterization**. The SEM images were taken with a FEI Apreo S LoVac scanning electron microscope. The TEM images were obtained with the JEOL-2100F system equipped with the EDAX Genesis XM2. XPS measurements were performed on a photoelectron spectrometer using Al $K\alpha$ radiation as the excitation source (PHI 5000 VersaProbe). All the peaks were calibrated with the C1s spectrum at a binding energy of 284.8 eV. The ex situ and in situ XRD were measured at the Rigaku Smartlab9KW Diffraction System using a Cu $K\alpha$ source ($\lambda = 0.15406$ nm). The in situ Raman spectroscopy was acquired on a Renishaw inVia reflex Raman microscope under an excitation of 532 nm laser. The XAS measurements were undertaken at the 1W1B beamline of the Beijing Synchrotron Radiation Facility (BSRF). The XAS spectra were analyzed with the ATHENA software package. To exclude the interference of Cu substrate, CuS powders were stripped off from the Cu substrate and loaded on CP for in situ XRD, in situ Raman, and in situ XANES tests. The nuclear magnetic resonance (NMR) spectra were recorded on Varian Mercury Plus 400 instruments at 400 MHz ($^1$H NMR) and 101 MHz ($^{13}$C NMR). Chemical shifts were reported in parts per million downfield from internal tetra-methylsilane. Multiplicity was indicated as follows: s (singlet), d (doublet), m (multiplet). Coupling constants were reported in hertz (Hz). The gas chromatograph (GC)-mass spectrometry was carried out with TRACE DSQ. The GC was measured on Agilent 7890A with thermal conductivity and flame ionization detector. The injection temperature was set at 300 °C. Nitrogen was used as the carrier gas at 1.5 mL min$^{-1}$. The high-performance liquid chromatography was measured on the Agilent 1220.

**Electrochemical measurements**. Electrochemical measurements were carried out in a divided three-compartment electrochemical cell consisting of a working electrode, a carbon rod counter-electrode, and a Hg/HgO reference electrode. The cathode cell (10 mL) and anode cell (10 mL) containing 1.0 M KOH solution (2.0 mL Diox and 5.0 mL H$_2$O), respectively, were separated by the membrane. After the Cu-S NSs in situ formed, 0.1 mmol of alkynes dissolved in 2.0 ml Diox were rapidly added into the cathode and stirred to form a homogeneous solution. Then, chronoamperometry was carried out at a given constant potential $-1.3$ V vs. Hg/HgO and stirred until the starting substrates disappeared. After that, the products at the cathode were extracted with dichloromethane (DCM). The DCM phase was removed and the residuals were subjected to be separated either by flash column chromatography or using a thin-layer chromatography plate to give the isolated yields or was analyzed by GC to provide the GC conversion yields. The GC yields were calculated according to standard calibration curves. The procedure and reaction setup of electrochemical semideuteration of alkynes was similar to the semi-hydrogenation process, except that the H$_2$O and 1.0 M KOH were replaced by D$_2$O and 0.5 M K$_2$CO$_3$, respectively. The procedure and reaction setup of electrochemical semi-hydrogenation of alkynes over Cu NAs were similar to that of Cu-S NSs, except that the working electrode changed to Cu NAs.

The procedure for the hydrodehalogenation of 2-iodobenzyl alcohol was similar to that of the alkyne semi-hydrogenation. Typically, 0.2 mmol of 2-iodobenzyl alcohol was added into the cathode cell. The LSV curves were recorded voltage range $-0.9 \sim -1.5$ V vs. Hg/HgO at a scan rate of 5 mV s$^{-1}$ and chronoamperometry was carried out at a given constant potential $-1.3$ V vs. Hg/HgO for 1 h using in situ-formed Cu NAs as cathode with and without adding 0.05 mmol of Na$_2$S, respectively. After adding Na$_2$S, the current density increased obviously and the conversion of 2-iodobenzyl alcohol improved from 36% to 99%, demonstrating the general applicability of our treating method.

**Quantitative reductive product**. The conversion (%), selectivity (%), and Faradic efficiency (FE, %) of alkenes, as well as the deuterated ratio (%) of deuterated alkenes were calculated using Eqs. (1)–(3):

$$\text{Conversion (\%)} = \frac{\text{mol of formed alkene}}{\text{mol of initial alkyne}} \times 100\% \qquad (1)$$

$$\text{Selectivity (\%)} = \frac{\text{mol of formed alkene}}{\text{mol of consumed alkyne}} \times 100\% \qquad (2)$$

$$\text{Deuteration ratio (\%)} = 100 - \left[ \left( \frac{\text{residual integral}}{\text{number of labeling sites}} \right) \times 100\% \right] \qquad (3)$$

*Electrochemical in situ XRD measurements*. The in situ electrochemical XRD pattern were measured at a Rigaku Smartlab9KW Diffraction System using a Cu $K\alpha$ source ($\lambda = 0.15406$ nm). The electrolytic cell was homemade by Teflon with Pt wire as the counter-electrode and Hg/HgO electrode as the reference electrode. The pattern was collected in the $2\theta$ ranging from 40.5° to 60.5° under the applied potential from without bias to $-1.3$ V vs. Hg/HgO. Each diffraction pattern was collected for 5 min for statistics.

*Electrochemical in situ Raman measurements*. The in situ electrochemical Raman spectroscopy was recorded on the Renishaw inVia reflex Raman microscope under an excitation of 532 nm laser under controlled potentials by an electrochemical workstation. The electrolytic cell was homemade by Teflon with a piece of round quartz glass as the cover to protect the objective. The working electrode was set to keep the plane of the sample perpendicular to the incident laser. Pt wire as the counter-electrode was rolled to a circle around the working electrode. Hg/HgO electrode with an internal reference electrolyte of 1.0 M KOH was used as the reference electrode. The spectrum was recorded every 30 s under the applied potential from without bias to $-1.3$ V vs. Hg/HgO.

*Electrochemical in situ XAS measurements*. The in situ electrochemical XAS at the Cu $K$-edge was recorded at at the 1W1B beamline of the BSRF. The electrolytic cell was homemade by Teflon with Pt plate as the counter-electrode and Hg/HgO electrode as the reference electrode. The applied potential was set at $-1.3$ V vs. Hg/HgO during the activation. The photon energy was calibrated with the first inflection point of the Cu K-edge in Cu metal foil. The XAS spectra were analyzed with the ATHENA software package.

**Computational details**. In this work, all calculations were performed using the Vienna ab initio simulation package based on the DFT[54,55]. Projector augmented wave pseudo-potentials were used[56]. The exchange-correlation contributions to the total energy were estimated by the generalized gradient approximation with Perdew–Burke–Ernzerhof form[57]. The Hubbard U approach (DFT+U) was adopted to better describe the on-site Coulomb correlation of the localized $3d$ electrons for Cu with U–J = 3.87 eV[58,59]. An empirical dispersion-corrected DFT

method (DFT-D3) was carried out to reasonably describe the weak long-distance van der Waals effects[60]. The kinetic energy cutoff for the plane-wave expansion was set to 500 eV. An energy convergence threshold of $10^{-4}$ eV was set in the self-consistent field iteration. The geometry optimization within the conjugate gradient method was performed with forces on each atom <0.05 eV Å$^{-1}$. A $p$ (2 × 2) slab model with three atomic layers was adopted to simulate the Cu (111) surface. A vacuum layer of 15 Å was inserted along the $c$ direction to eliminate the periodic image interactions. S-doped Cu surface was modeled with one surface Cu being replaced by an S atom. The bottom atomic layer was fixed, while other layers and the adsorbates were fully relaxed during structural optimizations. The Brillouin zone was sampled by a $k$-point mesh of 4 × 4 × 1. The reaction free energy change can be obtained with the following equation:

$$\Delta G = \Delta E + \Delta E_{ZPE} - T\Delta S$$

where $\Delta E$ is the total energy difference between the products and the reactants of each reaction step, and $\Delta E_{ZPE}$ and $\Delta S$ are the differences of zero-point energy and entropy, respectively. The zero-point energy of free molecules and adsorbates were obtained from the vibrational frequency calculations. The free energy change of each step that involves an electrochemical proton–electron transfer was described by the computational hydrogen electrode (CHE) model proposed by Nørskov et al.[61]. In this technique, zero voltage is defined based on the reversible hydrogen electrode, in which the reaction is defined to be in equilibrium at zero voltage, at all values of pH, at all temperatures, and with H$_2$ at 101,325 Pa pressure. Therefore, in the CHE model, the free energy of a proton–electron pair is equal to half of the free energy of gaseous hydrogen at a potential of 0 V.

## Data availability
The data that support other plots within this paper are available from the corresponding author upon reasonable request. Source data are provided with this paper.

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

## Acknowledgements

We acknowledge the National Natural Science Foundation of China (number 21871206) for support. We acknowledge Dr. Lirong Zheng and the 1W1B beamline of the Beijing Synchrotron Radiation Facility for supporting this project.

## Author contributions

B.Z. conceived the idea and directed the project. B.Z. and Y.W. designed the experiments. Y.W and C.L. carried out the experiments. C.W. performed and analyzed the DFT calculations. Y.S. and Y.W. carried out the in situ XRD measurements. Y.W. and C.L. wrote the paper. B.Z. revised the paper. All authors discussed the results and commented on the paper.

## Competing interests

The authors declare no competing interests.

## Additional information

**Peer review information** *Nature Communications* thanks the anonymous reviewers for their contribtuions to the peer review of this work. Peer review reports are available.

