## [Peer Review File · Nature Communications]

REVIEWER COMMENTS

Reviewer #1 (Remarks to the Author):

Zhang and co-workers detail the development of a novel electrocatalytic system for the semi hydrogenation of alkynes to cis alkenes. The electrocatalyst involves an in situ electroreduction of CuS to Cu with adsorbed sulfur on the surface. These catalysts are extremely well characterized by XRD, EDS, SEM, XPS, XAS, and their activity and the role of the surface sulfur was analyzed by voltammetry and Raman spectroscopy. Their electrocatalytic hydrogenation system was then tested on a small selection of alkynes and was found to give alkene to alkyne selectivity of 99% with good to excellent yields, in a time and potential independent manner. The manuscript is well written and it is a carefully put together piece of work.

While these novel catalysts are well characterized, the primary synthetic advance is an improvement in the selectivity from the >95:5 selectivity developed in their previous publication (ref 18: 10.1002/anie.202009757) to 99:1 in this manuscript. As the two products, alkene and alkane, should be readily separated by column chromatography, it's not obvious if this improvement is so profound. The mechanism experiments neatly outline the role of surface bound sulfur in rendering the over hydrogenation thermodynamically unfavourable. This latter point is key and provides the framework for the time and potential-independent nature of this catalyst system. As I am best able to comment on the synthetic aspects of this work, before recommending the manuscript to be accepted, I would like to see this aspect pushed further. Can the authors run a constant current experiment and then take aliquots with which to analyse and track the evolving selectivity, hence a chronopotentiometry trace can be plotted with selectivity on a second y-axis. For the time and potential independent feature to be of value, it must therefore be demonstrated and documented. Showing that the over hydrogenation is really very challenging will provide that evidence. Constant current (chronopotentiometry) is how synthetic chemists will want to run these reactions and therefore the authors also need to demonstrate the selectivity under these conditions and the product yields. If the selectivity remains as high and the catalyst system is as synthetically useful as they claim, then I can recommend publication in Nat Commun.

Reviewer #2 (Remarks to the Author):

The work presented in this manuscript employs the application of electrosynthesis to the hydrogenation of alkynes to alkenes. The authors present the reaction as well as a carefully planned production of a Cu-S nanowire sponge as catalyst. The characterization of the catalyst, the electrode and the efficacy of this setup for alkyne hydrogenation has been carried out to a high standard with broad appeal across the chemical sciences and worthy of publication. The only comments I make are in relation to the presentation of the manuscript, and recommend the authors attend to several instances of mismatched tenses and awkward word choices that make appreciating this work more arduous than it should. Firstly, there is an overusage of "in situ" throughout the first half of the manuscript. For example, there are 10 mentions on page 8 alone and the authors should consider "in situ" as the default setting such that it need not be mentioned so frequently. The authors can simply state when it is instead ex situ.

Line 42 - semi-hydrogenation of an alkyne will always give an alkene, so mentioning alkene is redundant.

Line 43 - "But, the good alkenes selectivity..." can be "Good alkene selectivity..."

Line 49 - "electrocatalysts for highly selective alkyne semi-hydrogenation..."

Line 59 - What is "oxides-reduced"?

Line 79 - "computational" instead of "calculation"

Fig. 1 - "sulfur" instead of "Sulfur"

Line 107 - "Pourbaix" is a person

Line 111 - "size of nanoparticles-based nanowires and cause the collapse of nanowire arrays to form nanowire sponges" is a word salad of the prefix nano, and must be revised, e.g. "the size of the nanowires and cause the conversion of arrays into sponges."

Line 120 - "the percentage of atomic sulfur"

Line 151 "electrochemical in situ Raman spectroscopy" and all other instances of "electrochemical in situ measurements" are better described as "spectroelectrochemical" measurements, which covers all the techniques presented here.

Line 152: the S-S bond vibration is 472 wavenumbers, and not nanometers. This error is also in Fig. 3b.

Line 230 - "direct electroduction of Cu(OH)₂ nanowire arrays were utilized" needs to be rephrased.

Line 267 - "The speculated cation/anion ion pair complex" should be "The proposed cation/anion pair complex"

Line 282 - "enhanced alkyne semi-hydrogenation"

Line 330 - "or probe molecules for studying reaction mechanisms."

Lines 343/373 - "hydrodehalogenation" by definition results in removing a halide, so the "of halide" is redundant.

Line 355 - The first two sentences of the Discussion sentence are poorly phrased and need to be rewritten.

Line 371 - delete "as the deuterated source"

Line 375 - delete "to alkenes" as it is implied

Line 376 - "low-cost, nanostructured electrodes"

The calculated thermodynamic values (free energy) should be expressed as kcal/mol or kJ/mol rather than eV as the former are the preferred units for bond/binding strength, and Fig. 6 modified accordingly.

The authors should consider showing the first derivative in the inset of Fig. 3c in order to show the variation in the rising edge of the Cu K-edge spectra.

The authors did not use sulfur K-edge XAS in addition to Cu K-edge XAS - is there a reason for this? The significantly lower energy, a higher resolution is offered by the S K-edge, and given the lower concentration of sulfur in the sample (cf. Cu) and its on or close to the surface, it would be more informative.

A point-by-point response to the reviewers' comments

To reviewer 1:

Reviewer letter: Zhang and co-workers detail the development of a novel electrocatalytic system for the semi hydrogenation of alkynes to cis alkenes. The electrocatalyst involves an in situ electroreduction of CuS to Cu with adsorbed sulfur on the surface. These catalysts are extremely well characterized by XRD, EDS, SEM, XPS, XAS, and their activity and the role of the surface sulfur was analyzed by voltammetry and Raman spectroscopy. Their electrocatalytic hydrogenation system was then tested on a small selection of alkynes and was found to give alkene to alkyne selectivity of 99% with good to excellent yields, in a time and potential independent manner. The manuscript is well written and it is a carefully put together piece of work.

While these novel catalysts are well characterized, the primary synthetic advance is an improvement in the selectivity from the >95:5 selectivity developed in their previous publication (ref 18: 10.1002/anie.202009757) to 99:1 in this manuscript. As the two products, alkene and alkane, should be readily separated by column chromatography, it is not obvious if this improvement is so profound. The mechanism experiments neatly outline the role of surface bound sulfur in rendering the over hydrogenation thermodynamically unfavourable. This latter point is key and provides the framework for the time and potential-independent nature of this catalyst system. As I am best able to comment on the synthetic aspects of this work, before recommending the manuscript to be accepted, I would like to see this aspect pushed further. Can the authors run a constant current experiment and then take aliquots with which to analyse and track the evolving selectivity, hence a chronopotentiometry trace can be plotted with selectivity on a second y-axis. For the time and potential independent feature to be of value, it must therefore be demonstrated and documented. Showing that the over hydrogenation is really very challenging will provide that evidence.

Constant current (chronopotentiometry) is how synthetic chemists will want to run these reactions and therefore the authors also need to demonstrate the selectivity under these conditions and the product yields.

If the selectivity remains as high and the catalyst system is as synthetically useful as they claim, then I can recommend publication in *Nat Commun*.

Answer: We highly appreciate the reviewer for the very positive comments and kind suggestions on our

manuscript. According to the kind concern, we provide a specific response. To save your and the reviewers' valuable time, key revisions are displayed in a yellow background in the revised manuscript and supporting information.

As for the main comment on "While these novel catalysts are well characterized, the primary synthetic advance is an improvement in the selectivity from the >95:5 selectivity developed in their previous publication (ref 18: 10.1002/anie.202009757) to 99:1 in this manuscript.", I would like to explain here. In our previous publications (Ref. 18), the selectivity is good but is highly potential- and time-dependent over Pd-P, a noble metal material. To obtain 95% selectivity, we need to control the reaction time and the applied potential carefully. Otherwise, the over-hydrogenated alkanes are the main products.

So, to solve these problems, we present a designed low-cost S-doped and -adsorbed Cu materials for potential-independent and time-independent semi-hydrogenation of alkynes. The lattice doped sulfur can weaken the alkene adsorption and make the further hydrogenation of alkene thermodynamically disadvantageous, determining the high selectivity of alkenes over Cu-S cathode. Thus, the novelty of this manuscript is designing a low-cost electrocatalyst for potential-independent and time-independent selective semi-hydrogenation of alkynes.

Figure R1. a Current-dependent and b time-dependent alkynes conversion yield (Con.) and alkenes selectivity (Sel.) over Cu-S NSs.

According to the reviewer's kind suggestion, constant current experiments are conducted, and the results showed in Figure R1 (Figure S11 in the Supporting Information). Over 98% alkene selectivity are maintained at the cathode continuous current at 10, 20, 30, and 40 mA for 4-hour reaction (Figure R1a

or Figure S11a in the Supporting Information), suggesting this current-independent selectivity over Cu-S NSs. And 99% alkynes conversion is achieved at 40 mA within 4h. Moreover, time-dependent curves (Figure R1b or Figure S11b in the Supporting Information) show that alkenes selectivity did not vary with reaction time, even after the complete disappearance of alkynes, suggesting the time-independent selectivity under constant current test. These satisfactory results show that the over hydrogenation problem that often appeared in reported references can be solved using Cu-S cathode in our manuscript, as the reviewer concerned.

To reviewer 2:

Reviewer letter: The work presented in this manuscript employs the application of electrosynthesis to the hydrogenation of alkynes to alkenes. The authors present the reaction as well as a carefully planned production of a Cu-S nanowire sponge as catalyst. The characterization of the catalyst, the electrode and the efficacy of this setup for alkyne hydrogenation has been carried out to a high standard with broad appeal across the chemical sciences and worthy of publication. The only comments I make are in relation to the presentation of the manuscript, and recommend the authors attend to several instances of mismatched tenses and awkward word choices that make appreciating this work more arduous than it should.

Answer: We deeply thank the reviewer for the very positive comments on our work and sincerely appreciate the careful and nice suggestions on our manuscript. According to the kind suggestions, we have tried our best to check and polish the language of this manuscript, and marked the modifications with yellow background. As for his/her concerns or confuse on our paper, we have given detailed response, respectively.

Comment 1: Firstly, there is an overusage of "in situ" throughout the first half of the manuscript. For example, there are 10 mentions on page 8 alone and the authors should consider "in situ" as the default setting such that it need not be mentioned so frequently. The authors can simply state when it is instead ex situ.

Answer: We sincerely acknowledge the reviewer for the kind suggestion. We have taken "in situ" as the default setting in the part of *in situ* characterization to make the expression more concise.

Comment 2: Line 42 - semi-hydrogenation of an alkyne will always give an alkene, so mentioning alkene is redundant.

Answer: We appreciate the reviewer for the kind suggestion. The "alkenes" in line 42 has been deleted.

Comment 3: Line 43 - "But, the good alkenes selectivity..." can be "Good alkene selectivity..."

Answer: Thanks for the reviewer's kind comment. We have replaced "But, the good alkenes selectivity..." by "Good alkene selectivity..." in line 43 and the modification displayed in a yellow background in the revised manuscript.

Comment 4: Line 49 - "electrocatalysts for highly selective alkyne semi-hydrogenation..."

Answer: Thanks for the reviewer's kind suggestion. The expression in line 49 has been revised.

Comment 5: Line 59 - What is "oxides-reduced"?

Answer: The "oxides-reduced Cu" in line 59 means Cu catalyst electroreduced from copper oxide. To make the expression clearer, "oxides-reduced Cu" was replaced by "copper oxide-derived Cu".

Comment 6: Line 79 - "computational" instead of "calculation"

Answer: The "calculation" has been replaced by "computational".

Comment 7: Fig. 1 - "sulfur" instead of "Sulfur"

Answer: The spelling mistake in Fig. 1 has been corrected.

Comment 8: Line 107 - "Pourbaix" is a person

Answer: The spelling mistake of "Pourbaix" has been corrected and displayed in a yellow background in the revised manuscript.

Comment 9: Line 111 - "size of nanoparticles-based nanowires and cause the collapse of nanowire arrays to form nanowire sponges" is a word salad of the prefix nano, and must be revised, e.g. "the size of the nanowires and cause the conversion of arrays into sponges."

Answer: Thanks for the reviewer's wise suggestion. The sentence in line 111 has been rephased as "*The electroreduction-induced sulfur stripping could decrease the size of the nanowires and cause the transformation of nanowire arrays to nanowire sponges*".

Comment 10: Line 120 - "the percentage of atomic sulfur"

Answer: Thanks for the reviewer's comment. The sulfur content mentioned in our manuscript is defined as the number of S atoms relative to the total number of S and Cu atoms, so it is atomic ratio rather than weight ratio. Thus, the expression of "*the atomic percentage of sulfur*" is to distinguish it with the "*the wet percentage of sulfur*".

Comment 11: Line 151 "electrochemical in situ Raman spectroscopy" and all other instances of "electrochemical in situ measurements" are better described as "spectroelectrochemical" measurements, which covers all the techniques presented here.

Answer: Thanks for the reviewer's kind suggestion. In the description of characterization methods and results, we use "electrochemical in situ Raman spectroscopy" and "X-ray absorption near-edge structure spectra" to avoid confusion and make the expression clearer. And in other general statements, such as "Abstracts" and "Discussion" portions, we use "*ex situ* and *in situ* results" covers all the techniques to make the expression more concise.

Comment 12: Line 152: the S-S bond vibration is 472 wavenumbers, and not nanometers. This error is also in Fig. 3b.

Answer: The mistake in Fig. 3b has been corrected and displayed in a yellow background in the revised manuscript.

Comment 13: Line 230 - "direct electroreduction of Cu(OH)₂ nanowire arrays were utilized" needs to be rephrased.

Answer: This sentence was rephrased as "Cu nanowire arrays (donate as Cu NAs) were synthesized by *in situ* electroreduction of Cu(OH)₂ NAs."

Comment 14: Line 267 - "The speculated cation/anion ion pair complex" should be "The proposed cation/anion pair complex"

Answer: "The speculated cation/anion ion pair complex" in line 267 has been replaced by "The proposed cation/anion pair complex" and displayed in a yellow background in the revised manuscript.

Comment 15: Line 282 - "enhanced alkyne semi-hydrogenation"

Answer: The "enhanced the alkynes semi-hydrogenation" in line 282 has been replaced by "enhanced the alkynes semi-hydrogenation activity" and displayed in a yellow background in the revised manuscript.

Comment 16: Line 330 - "or probe molecules for studying reaction mechanisms."

Answer: The "or the probe molecules for studying reaction mechanisms" has been modified as "or probe molecules for studying reaction mechanisms".

Comment 17: Lines 343/373 - "hydrodehalogenation" by definition results in remvng a halide, so the "of halide" is redundant

Answer: The "of halide" in line 343 and line 373 have been deleted.

Comment 18: Line 355 - The first two sentences of the Discussion sentence are poorly phrased and need to be re-written.

Answer: The first two sentences of the Discussion have been rephrased and displayed in a yellow background in the revised manuscript.

Comment 19: Line 371 - delete "as the deuterated source"

Answer: The "as the deuterated source" in line 371 has been deleted in the revised manuscript.

Comment 20: Line 375 - delete "to alkenes" as it is implied

Answer: The "to alkenes" in line 375 has been deleted in the revised manuscript.

Comment 21: Line 376 - "low-cost, nanostructured electrodes"

Answer: The expression of "low-cost nanostructured electrodes" in line 376 has been modified to "low-cost, nanostructured electrodes" in the revised manuscript.

Comment 22: The calculated thermodynamic values (free energy) should be expressed as kcal/mol or kJ/mol rather than eV as the former are the preferred units for bond/binding strength, and Fig. 6 modified accordingly.

Answer: Thanks for the reviewer's kind suggestion. The kilojoule (KJ/mol) is a SI (International System of Units) derived unit of energy per amount of material. As the reviewer mentioned, this unit is usually used to quantify the Gibbs free energy and describe the bond/binding strength. The electronvolt (eV) is a unit of energy used in physics. Energy in electronvolts is equal to the electric charge Q in elementary charge times the potential difference V in volts. This unit can be used in the theoretical electrochemical calculations to conveniently relate the thermodynamic reaction free energy to the applied external voltage. We can convert eV to KJ/mol by $1 \text{ eV} = 96.487 \text{ KJ/mol}$. So, these two units can be used interchangeably from different point of view. Here, according to the reviewer's kind suggestion, we change the energy unit of eV to KJ/mol in the revised manuscript and supporting information.

Comment 23: The authors should consider showing the first derivative in the inset of Fig. 3c in order to show the variation in the rising edge of the Cu K-edge spectra.

Answer: Thanks for the reviewer's kind suggestion. According to the nice suggestion, the first derivative of the XANES spectra for Cu-S NSs at -1.3 V vs. Hg/HgO, Cu foil, CuS, and Cu₂S were displayed in Figure R2 or Figure S7 in the supporting information. It is clear that the first-order derivative of the XANES

spectra for Cu-S NSs at -1.3 V is dominated by the Cu(0) feature, consistent with the result in Fig. 3, and further proved the successful synthesis of low-coordinated Cu with surface sulfur doping and adsorption.

Figure R2. First-order derivatives of the XANES spectra for Cu-S NSs at -1.3 V vs. Hg/HgO, Cu foil, CuS, and Cu₂S.

Comment 24: The authors did not use sulfur K-edge XAS in addition to Cu K-edge XAS - is there a reason for this? The significantly lower energy, a higher resolution is offered by the S K-edge, and given the lower concentration of sulfur in the sample (cf. Cu) and its on or close to the surface, it would be more informative.

Answer: Thanks for the reviewer's kind suggestion. We strongly agree the reviewer's opinion that sulfur K-edge XAS will offer more information with higher precision and resolution. However, the hard X-ray beamline we conducted the Cu K-edge XAS is not workable for S K-edge XAS. Because the X-ray photon energy range for Cu K-edge XAS is 8950~9200 eV, while 2440~2560 eV for S K-edge. In addition, both experimental and theoretical results have indicated that Cu is the active site in our reaction system, so we focused more on the real valence and coordination environment of Cu during the reaction process, thus unveiling the origin of high activity and selectivity.

We highly appreciate all the reviewers' thorough reading and comments/questions about our manuscript! We are sure that the quality of this work will be greatly improved according to these nice comments and wise suggestions.

REVIEWER COMMENTS

Reviewer #1 (Remarks to the Author):

The request for the constant current experiment was performed and showed high selectivity throughout, which is impressive. However, my request was to use constant current but then plot the applied potential on one y-axis and selectivity/yield on a separate y-axis (with time on the x axis). The authors then should apply more charge than is needed (for example to apply 30 mA for 10-12 hours) to understand at what point does the selectivity start to diminish, e.g., at what applied potential, at what time and/or amount of charge passed does it diminish. Presumably the selectivity will start to diminish at some point, but showing when it diminishes will be really valuable indeed. I still think this would be good to see before recommending publication.

A point-by-point response to the reviewers' comments

To reviewer:

Reviewer letter: The request for the constant current experiment was performed and showed high selectivity throughout, which is impressive. However, my request was to use constant current but then plot the applied potential on one y-axis and selectivity/yield on a separate y-axis (with time on the x axis). The authors then should apply more charge than is needed (for example to apply 30 mA for 10-12 hours) to understand at what point does the selectivity start to diminish, e.g., at what applied potential, at what time and/or amount of charge passed does it diminish. Presumably the selectivity will start to diminish at some point, but showing when it diminishes will be really valuable indeed.

I still think this would be good to see before recommending publication.

Answer: We highly appreciate the reviewer for his/her wise comments. According to the reviewer's constructive suggestions, we have carried out the long-time electrolysis experiments under a constant current of -30 mA over the in situ formed Cu-S NSs cathode. As shown in Figure R1a, the model substrate 4-ethynylaniline (**1a**) could be fully consumed within 5 h to generate the alkene **2a** with 98% selectivity. No selectivity diminish of **2a** was observed even if the reaction time was prolonged to 18 h. This result illustrates that the excellent alkene selectivity is irrelevant to the input potential/current and the reaction time, highlighting the intrinsic control by the Cu-S NSs cathode. In addition, only approximately 30 mV (from -1.37 to -1.4 V) of the potential (E) increased during 18 h, which might be due to the hydrogen evolution reaction (HER) that gradually became dominant with alkynes substrate consumption.

To further verify the time-independent selectivity of alkene, alkene **2a** as the starting material was subjected to our reaction system under identical conditions with the reaction of alkyne **1a**. As expected, no alkane product was detected during a 7 h reaction under a constant current of -30 mA. This further rationalized our speculation on Cu-S NSs cathode controlled excellent selectivity of alkene (Figure R1b).

Therefore, our work not only provides a new understanding of the activity and selectivity origin of sulfur modified transition metal toward highly selective semi-hydrogenation of alkynes, but also offers a paradigm for designing and synthesizing low-cost, nanostructured electrode via an *in situ* electrochemical treatment for other electrocatalytic hydrogenation reactions.

These results are also included in the revised manuscript and Supplementary Information (Figure S11). To save the reviewer's valuable time, key revisions are displayed in a yellow background in the

revised manuscript and Supplementary Information.

Figure R1. **a** Time-dependent conversion yield (Con.) of 4-ethynylaniline (**1a**), alkene (**2a**) selectivity (Sel.), and potential (E) variations over an in situ formed Cu-S NSs cathode. **b** Time-dependent alkene (**2a**) conversion yield (Con.) and potential (E) variations over an in situ formed Cu-S NSs cathode.

We highly appreciate the reviewer's thorough reading and wise comments about our manuscript! We are sure that the quality of this work will be greatly improved according to these nice comments and wise suggestions.

REVIEWERS' COMMENTS

Reviewer #1 (Remarks to the Author):

the authors have now completed the recommended experiment and have shown that indeed, it is a highly selective catalyst system. I commend the authors on a very nice piece of work and I can now recommended publication.

Department of Chemistry
Tianjin University
Tianjin 300072, P. R. China
Tel&Fax: 86-22-27403475
E-mail: bzhang@tju.edu.cn

A point-by-point response to the reviewers' comments

To reviewer:

Reviewer letter: The authors have now completed the recommended experiment and have shown that indeed, it is a highly selective catalyst system. I commend the authors on a very nice piece of work and I can now recommended publication.

Answer: We highly appreciate the reviewer for his/her positive comments on our revised manuscript. We are sure that the quality of this work has been greatly improved according to these nice comments and wise suggestions. Thanks very much.